# Effect of Invasion of *Borrelia burgdorferi* in Normal and Neoplastic Mammary Epithelial Cells

**DOI:** 10.3390/antibiotics10111295

**Published:** 2021-10-24

**Authors:** Gauri Gaur, Janhavi Y. Sawant, Ankita S. Chavan, Vishwa A. Khatri, Yueh-Hsin Liu, Min Zhang, Eva Sapi

**Affiliations:** 1Lyme Disease Research Group, Department of Biology and Environmental Science, University of New Haven, 300 Boston Post Road, GH 104A, West Haven, CT 06516, USA; ggaur2@unh.newhaven.edu (G.G.); jsawa1@unh.newhaven.edu (J.Y.S.); achav6@unh.newhaven.edu (A.S.C.); vkhat1@unh.newhaven.edu (V.A.K.); yliu10@unh.newhaven.edu (Y.-H.L.); mzhan7@unh.newhaven.edu (M.Z.); 2Department of Criminal Justice, Coppin State University, Baltimore, MD 21216, USA

**Keywords:** Lyme disease, breast cancer, invasion, EMT markers

## Abstract

*Borrelia burgdorferi*, the causative agent of Lyme Disease, is known to be able to disseminate and colonize various organs and tissues of its hosts, which is very crucial for its pathogenicity and survival. Recent studies have shown the presence of *B. burgdorferi* DNA in various breast cancer tissues, in some with poor prognosis, which raises the question about whether *B. burgdorferi* can interact with mammary epithelial cells and could have any effect on their physiology, including tumorigenic processes. As the model in this study, we have used MCF 10A normal and MDA-MB-231 tumorigenic mammary epithelial cells and infected both cell lines with *B. burgdorferi*. Our immunofluorescence and confocal microscopy results showed that *B. burgdorferi* is capable of invading normal epithelial and breast carcinoma cell lines within 24 h; however, the infection rate for the breast carcinoma cell lines was significantly higher. While the infection of epithelial cells with *B. burgdorferi* did not cause any changes in cell proliferation rates, it showed a significant effect on the invasion and migratory capacity of the breast cancer cells, but not on the normal epithelial cells, as determined by Matrigel invasion and wound healing assays. We have also found that the levels of expression of several epithelial–mesenchymal transition (EMT) markers (fibronectin, vimentin, and Twist1/2) changed, with a significant increase in tissue remodeling marker (MMP-9) in MDA-MB-231 cells demonstrated by quantitative Western blot analyses. This observation further confirmed that *B. burgdorferi* infection can affect the in vitro migratory and invasive properties of MDA-MB-231 tumorigenic mammary epithelial cells. In summary, our results suggest that *B. burgdorferi* can invade breast cancer tumor cells and it can increase their tumorigenic phenotype, which urges the need for further studies on whether *B. burgdorferi* could have any role in breast cancer development.

## 1. Introduction

*Borrelia burgdorferi* sensu lato is the causative agent of Lyme Disease, an infectious disease spreading across the United States, Europe, Asia, Australia, as well as some parts of Africa and South America [1]. *B. burgdorferi* is well known to be able to disseminate and colonize various organs and tissues of its natural reservoir hosts, which is a very crucial step for its pathogenicity and overall survival [2]. To be able to invade and persist in the different tissues, *B. burgdorferi* has developed several very sophisticated strategies, including manipulating the innate and adaptive immune systems of the host, binding to host proteins to initiate vascular and extracellular matrix interactions, even high-jacking and stimulating host proteases to digest and remodel the healthy tissues [3]. Intracellular localization of *B. burgdorferi* was also reported for different mammalian cells, which can help the bacterium to evade the host’s immune system and other environmental stress factors [4] such as fibroblasts [5,6], neuronal and glial cells [7], endothelial cells [8], and macrophages [9]. It was also demonstrated that *B. burgdorferi* is able to adhere in vitro to epithelial cells in a time and dose-dependent fashion by attaching to the exposed cell surface proteoglycans [10]. However, the intracellular localization of *B. burgdorferi* in epithelial cells has not yet been studied. 

A recent microbiome study of breast cancer tissues detected *B. burgdorferi* DNA in several types of breast cancer tissues. In addition to this, the presence of *B. burgdorferi* in epidermal growth factor receptor 2 (HER2) positive tissues corresponded with a severe outcome [11]. The existence of *B. burgdorferi* in breast cancer tissues raised the question of whether this spirochete can colonize normal and tumorigenic breast epithelial cells and, if so, whether there is a difference in the kinetics of invasion.

Studies have shown that the invasion of host cells by bacteria can affect the host cell integrity due to manipulation of different molecular pathways, which in turn leads to cancer formation [12]. Consequently, once invaded, the bacteria have been shown to cause cell growth regulation which, in turn, leads to metastasis, increasing invasiveness and successively epithelial–mesenchymal transition (EMT) [13,14,15]. EMT is the phenomenon where the epithelial cells lose their cell-cell adhesion and polarity and attain mesenchymal properties, including motility and invasiveness [16].

Therefore, the goal of this study was to evaluate whether *B. burgdorferi* can invade epithelial cells derived from normal and breast carcinoma tissues cell lines using *B. burgdorferi* specific immunofluorescence staining and confocal microscopy methods. It further aimed to evaluate the rate at which *B. burgdorferi* invades breast epithelial cells by quantifying the number of spirochetes inside the cells. The second goal was to evaluate whether the invasion of *B. burgdorferi* in breast epithelial cells can cause in vitro tumorigenic changes in normal and breast carcinoma cell lines using various cellular and molecular biology techniques. Changes in cell viability, invasion, migration as well as epithelial–mesenchymal transition were assessed in this study.

## 2. Results

### 2.1. Infection of Breast Epithelial Cells with B. burgdorferi

The first part of the study was aimed to investigate the potential invasion of breast epithelial cells by *B. burgdorferi* using immunofluorescence staining techniques combined with confocal microscopy using the Leica SP8 confocal microscope. Z-stacks were generated utilizing Leica Application Suite X software. 

Normal (MCF 10A) and breast cancer epithelial cells (MDA-MB-231) were used as the in vitro model system for this study. MCF 10A, a human non-tumorigenic, normal epithelial cell line, has been utilized as normal mammary epithelial cells in numerous studies [17,18]. For breast cancer cells, a highly invasive, triple-negative MDA-MB-231 (ER^−^, PR^−^, HER2^−^) breast adenocarcinoma cell line was used [19,20].

The initial experiments consisted of finding the optimal infection ratio between *B. burgdorferi* and breast epithelial cells. Breast epithelial cells at 70% confluency were infected with *B. burgdorferi* at different MOIs using co-culture media (see Material and Methods) and an MOI of 60 was found to be the most optimum condition (data not shown).

The kinetics of invasion of *B. burgdorferi* in breast epithelial cells was studied by infecting the different cell lines for four different time points (4, 24, 48, and 72 h). After 4 h, the spirochetes were attached to the surface of the epithelial cells but did not invade the cells. Therefore, the subsequent experiments were conducted for 24, 48, and 72 h timepoints. 

The analysis of the longer time frames showed that the spirochetes are capable of attaching to the membrane or even entering a few of the normal epithelial cells within 24 h of infection (Figure 1, Panel I, shown by white arrowheads). The numbers of MCF 10A cells that were infected within 24 h with *B. burgdorferi* was on an average 36%, localized mostly near the membrane of the cell and also near the nucleus (Table 1, Figure 1, Panel IB). After 48 h of infection, 52% of the MCF 10A cells were infected and spirochetes were located near the nucleus (Figure 1, Panel IC, Table 1). Some small aggregates had also formed inside the cells (Panel IC, white arrow). After 72 h, the spirochetes that entered the cells were located near the nucleus of the cell as well, with about 61% of the cells invaded (Figure 1, Panel ID, Table 1). On comparing the infected and non-infected cells, there was no significant change in morphology seen after infection in MCF 10A cells (Figure 1). 

*B. burgdorferi* was also shown to be able to attach and invade MDA-MB-231 cells within 24 h and localize in the cytoplasm (Figure 1, Panel II). For MDA-MB-231, on average, about 89% of cells were invaded by *B. burgdorferi* spirochetes within 24 h while 92% of spirochetes invaded the mammalian cells at 48 h and 96% at 72 h (Table 1, Figure 1, Panel II B-D). Some of the spirochetes clearly formed small aggregates after 48 h of infection (Figure 1, Panel IIC, white arrow) which was also demonstrated by an enhanced confocal image near the cell nucleus (Figure 2, see also the Appendix A). However, 72 h after infection, most of the invading spirochetes were located in the cytoplasm, and some spirochetes were close to the nucleus (Figure 1, Panel II). Comparing the significance of the invasion of *B. burgdorferi* spirochetes into MDA-MB-231 and the normal MCF 10A cells, significant differences (*p*-values < 0.05) were found at each time point.

### 2.2. Effect of B. burgdorferi on the Cell Viability of Breast Epithelial Cells

The goal of the second set of experiments was to determine the effect of *B. burgdorferi* infection on tumorigenic phenotypes such as cell viability, migration, invasion, and epithelial–mesenchymal transition (EMT) in human epithelial cells using proliferation, migration, Matrigel based in vitro invasion assay, and quantitative Western blot analyses respectively.

The cell viability of MCF 10A and MDA-MB-231 cells was assessed after a 72 h infection with *B. burgdorferi* using standard cell proliferation assay (see Materials and Methods). The growth rate of the infected cells was compared to the uninfected epithelial cells to determine whether *B. burgdorferi* infection could change the proliferation rate of the cells. There was no significant change in cell viability observed for either MCF 10A or MDA-MB-231 cell lines as compared to the uninfected cells (Figure 3). 

### 2.3. Effect of B. burgdorferi on the Invasiveness of Breast Epithelial Cells

Subsequently, the invasiveness of the *B. burgdorferi* infected epithelial cells was analyzed using a Matrigel-based in vitro invasion assay and was compared to uninfected cells. *B. burgdorferi* infected MCF 10A showed no significant changes in the invasiveness as compared to the uninfected cells after 72 h (Figure 4 and Figure 5, Panel A and B). On the other hand, MDA-MB-231 showed an approximately 4-fold significant increase in invasiveness after infection with *B. burgdorferi*, as compared to the uninfected cells (Figure 4 and Figure 5, Panel C and D). 

### 2.4. Effect of B. burgdorferi on the Motility of Breast Epithelial Cells

The migration rates of the *B. burgdorferi*-infected epithelial cells were analyzed using a standard wound healing assay [21] and compared to uninfected cells. The highest change in the rate of migration was observed for the 48 h *B. burgdorferi* infected MCF 10A and MDA-MB-231 cells, 6 h after the wound was induced (Figure 6). For MCF 10A cells, the 48 h *B. burgdorferi* infected cells showed an approximate 10% increase in the rate of migration after 6 h of inducing the wound, while the non-infected MCF 10A cells also showed an approximate 10% increase in migration to close the wound (Figure 6, Panel I, Figure 7). On the other hand, the 48 h *B. burgdorferi* infected MDA-MB-231 showed a significant 48% wound recovery (*p*-value < 0.05) while the uninfected cells showed a 27% change in the rate of migration after 6 h of inducing the wound (Figure 6, Panel II, Figure 7).

### 2.5. Effect of B. burgdorferi on the EMT Markers of Breast Epithelial Cells

To further investigate the effect of *B. burgdorferi* in inducing EMT in normal and tumorigenic breast epithelial cells, the expression of common EMT-inducing transcription factors like fibronectin, E-Cadherin, N-Cadherin, MMP-9, Vimentin, and Twist1/2 with GAPDH as the loading control was analyzed using Western blot analyses. The protein from uninfected and 24, 48, and 72-h *B. burgdorferi* infected MCF 10A and MDA-MB-231 cells was analyzed using Western blot (see Section 4). 

The results show that the protein expression of several EMT markers increases in MDA-MB-231 infected with *B. burgdorferi* after 72 h as compared to the uninfected cells (Figure 8, right panel). There was a ~2-fold increase in fibronectin, activated MMP9, vimentin, Twist 1/2 protein expressions at 72 h of *B. burgdorferi* infection in MDA-MB-231 cells as compared to the uninfected cells. Quantification and statistical analyses of the expression levels of target proteins such as fibronectin, activated MMP9, vimentin, and Twist 1/2 from three independent experiments (see Section 4) revealed that change in activated MMP9 protein expression after 72 h of post-infection was indeed statistically significant (*p*-value < 0.05) in MDA-MB-231 cells (Figure 9, Panel A). Infected MCF 10A cells did not show any detectable changes for the expression of MMP-9, vimentin, and Twist 1/2 proteins as compared to the uninfected cells but an approximately 2-fold increase for fibronectin by 24 h of post-infection (Figure 8, left panel). However, after quantification and statistical analyses of this result from three independent experiments, it was not statistically significant (*p*-value > 0.05, Figure 9, Panel B).

Interestingly, there were no detectable changes in E-Cadherin and N-Cadherin protein expression for both *B. burgdorferi* infected MCF 10A and MDA-MB-231 cells as compared to their respective uninfected cells. 

## 3. Discussion

This study investigated the ability of *B. burgdorferi* to enter mammary epithelial cells and the potential effect of the infection on the physiology of the cells. Results showed that *B. burgdorferi* was indeed able to invade both normal and breast cancer epithelial cells within 24 h but at different rates. *B. burgdorferi* is able to attach and invade the cell membrane of the normal epithelial cells within 24 h, but the number of infected cells is significantly lower as compared to breast cancer cells even after 72 h. On the contrary, breast adenocarcinoma cells showed intracellular localization within 24 h of infection with *B. burgdorferi*, with over 80% of the cells showing the presence of the spirochetes. After 48 and 72 h, however, significantly more cancer cells were invaded by the *B. burgdorferi* spirochetes as compared to normal cells, and the spirochetes and small aggregates were located mostly near the nucleus of the cells. The ability of *B. burgdorferi* to invade cancer cells at a higher rate than the normal epithelial cells suggests that there is a different interaction between the cancer cells and the pathogen. It is well known that *B. burgdorferi* is capable of binding to membrane proteoglycans of the host cells [10]. Therefore, one possible explanation could be the different proteoglycan structures of cancer cells [22]. Modifications of the membrane glycan structure have been observed in malignant cells and have been proposed to be, in part, a driving force behind several biological processes of carcinogenesis [23]. Moreover, the gene expression of glycan-related genes has been shown to be very different in breast carcinomas compared to non-malignant tissue of the breast. This leads us to speculate that the altered extracellular matrix (ECM) of the cancer cells could promote a higher rate of invasion of *B. burgdorferi* as compared to the normal epithelial cells. Additionally, bacteria preferentially accumulate and proliferate within tumors due to the low oxygen concentration, which is attractive to *B. burgdorferi*, coupled with unique biochemical structures such as special receptors or metabolites, as well as the immune-privileged microenvironment for bacterial colonization [24].

Similar studies using differential immunofluorescence staining and microscopy have shown that *H. pylori* internalization occurred within 45 min of bacterial attaching to the surface of gastric epithelial cells [25]. Other studies showed that *H. pylori* could proliferate after entering the cells, with the maximum number of bacterial cells observed after 6–12 h. The ability of *B. burgdorferi* to invade breast epithelial cells within 24 h, as shown in this study, agrees with the pattern of invasion by other cancer-causing bacteria [26]. 

Other in vitro studies have also demonstrated the internalization of *B. burgdorferi* in various mammalian cells. For example, co-culturing human skin fibroblasts cells with *B. burgdorferi* at a MOI of 1000 showed *B. burgdorferi* binding to the fibroblast membrane after 24 h and invaginating within the fibroblast’s cytoplasm within 48 h [27]. *B. burgdorferi* spirochetes have been shown to attach to human synovial cells (SCs) along with partial or complete internalization into the cells after 90 min [28]. The spirochetes were mostly observed within the cytosol in the vicinity of the cytoplasmic filaments. Similarly, macrophages from peripheral blood and tissues showed a very early internalization of *B. burgdorferi*, but the spirochetes were destroyed thereafter. *B. burgdorferi* spirochetes were found to be coiled up inside the cytosol followed by signs of degradation within 90 min in the lysosomes at MOI of 100 [8,9]. This is not surprising as the function of the macrophages is to detect, engulf and destroy the pathogen as compared to fibroblast and epithelial cells, which are physical barriers for pathogens. Experiments have also shown that 10–25% of *B. burgdorferi* is able to intercellularly localize umbilical vein endothelial cells in vitro at ratios ranging from 200:1 to 500:1 within 24 h of infection [8]. *B. burgdorferi* has also been observed to start interacting with neuroglial cells within the first hour of co-infection at an MOI of 40 and seems to increase till after 20 h of co-incubation with neuronal and glial cells following an antibiotic challenge [7,29]. Another study has also shown *B. burgdorferi* (297 strain) to attach to and invade epithelial cells in vitro within 4 h of co-incubation at an MOI of 300 in a temperature-dependent manner [10]. The faster rate of invasion into the epithelial cells as compared to our study could be attributed to the use of a higher MOI and a more virulent strain of *B. burgdorferi*. The ability of *B. burgdorferi* to invade breast epithelial cells within 24 h as shown in this study, agrees with previous studies where the invasion in various cell types is faster than 24 h.

This study also addressed the effect of *B. burgdorferi* infection on the physiological changes of the infected mammary epithelial cells. Interestingly, MCF 10A and MDA-MB-231 cell lines did not show any significant change in cell proliferation rates after being infected with *B. burgdorferi* as compared to the uninfected cells. Similarly, no observable changes in cell viability were observed even after 7 days of co-culturing of *B. burgdorferi* with neuronal and glial cell lines [7,29]. A similar study further confirmed no effect on cell proliferation in neuroblastoma cells [30]. 

But the question remains—why do we see *B. burgdorferi* in breast cancer cells at such a high rate? Does *B. burgdorferi* serve any function in the breast cancer epithelial cells or was it just hiding from the immune system? To answer this question, tumorigenic-specific assays were also performed to evaluate any changes in cellular invasion and migration rate. 

Matrigel invasion assay was performed to compare the ability of the *B. burgdorferi* infected cells to invade the basement membrane as compared to the uninfected cells. Results shed light on the ability of *B. burgdorferi* to invade the surrounding tissues, colonization of the targeted tissue, and immune evasion, all of which are important steps for the pathogenesis of Lyme disease and cancer. While the invasive capacity of MCF 10A through a Matrigel did not change after infection, the infected MDA-MB-231 cells increased their invasiveness capacity by approximately 4-fold compared to the untreated cells. A wound-healing assay was performed to assess cell migration of infected versus uninfected cells, as modeling of cancer migration after *B. burgdorferi* infection. The results demonstrated a rapid rate of gap closure for the 48 h *B. burgdorferi* infected MDA-MB-231 cells after 6 h of inducing the wound, but no change was observed in the normal MCF 10A cells as compared to their uninfected controls. These obtained data strongly suggest that *B. burgdorferi* is capable of enhancing the rate of cell migration and invasion of cancer cells upon intracellular infection. The mechanism for such changes has been investigated for other pathogens. For example, some previous studies have demonstrated that *H. pylori* enhanced the rate of migration and invasion of human gastric cancer cells in vitro via various mechanisms such as calcium dependence and increasing heparinase expression [31]. A significant increase in the expression of Matrix metalloproteinases-9 (MMP-9) was also observed [31,32]. MMPs are enzymes that degrade the ECM and are considered to be important factors in facilitating tumor invasion and spread. Previous studies have shown that *B. burgdorferi* also induces MMP-9, utilizing it as an invasion strategy [4]. MMP-9 has been shown to play a potentially essential role in the dissemination of *B. burgdorferi* as well as play a role in the development of some of the signature manifestations of Lyme disease, including arthritis and carditis [33]. Another study has determined peptidoglycan (PGN) from the infectious bacterium *Staphylococcus aureus* (PGN-SA) led to the activation of Toll-like receptor 2 (TLR2) in MDA-MB-231, leading to an increase of invasiveness and adhesiveness of the cancer cells in vitro [34].

Lastly, this study evaluated the effect of *B. burgdorferi* invasion in normal and tumorigenic breast epithelial cells on the protein expression of common EMT biomarkers. The hallmark of EMT is usually the downregulation of epithelial markers like cell surface protein E-cadherin and the upregulation of mesenchymal markers like surface protein N-cadherin, cytoskeleton protein vimentin, and the transcription factor twist [35]. Additionally, compared with normal adult breast tissue lacking fibronectin, the upregulation of another EMT marker, extracellular matrix fibronectin, is associated with breast tumors [36]. The results of this study demonstrated a moderate increase in the expression of fibronectin protein in *B. burgdorferi* infected MCF 10A cells, but there was a detectable increase in the expression level of fibronectin, vimentin, and Twist1/2 protein and a statistically significant increase in activated MMP9 level at 72 h post *B. burgdorferi* infection in the MDA-MB-231 cells. In good agreement with our findings, a study that investigated the expression of some EMT markers during breast cancer progression also demonstrated vimentin expression positivity associated with aggressive tumors such as the triple-negative subtype [37]. Similarly, *H. pylori* significantly upregulates vimentin in the gastric cancer cell lines as well as increases the expression of MMP-7 [38]. Another study shows that infection of *Porphyromonas gingivalis* increases the aggressiveness, metastasis, and viability of oral squamous cell carcinoma (OSCC) through the induction of canonical EMT markers, MMP-9 and vimentin [14]. 

In summary, our studies show that *B. burgdorferi* is able to invade breast epithelial cells and its infection can increase the migration and invasion of cancer cells as well as increase the expression of MMP9. Further research in this area will illuminate the pathogenic mechanisms and survival mechanisms of *B. burgdorferi*. Additional research could identify potential therapeutic targets by the use of antibiotic therapy for cancer treatment. There has been a finding that shows that doxycycline, the frontline antibiotic treatment for Lyme Disease, in combination with Vitamin C, can eradicate cancer stem cells (CSCs) [39]. Other studies have shown that tigecycline, an antibiotic previously used for Lyme Disease treatment, alone or in combination with other chemotherapeutic drugs, is also a potential candidate for cancer treatment [40]. 

If we could provide more evidence that *B. burgdorferi* indeed can be present in metastatic tumors and could increase the invasion of breast cancer cells, it would give us means to develop therapeutic strategies with antibiotics as neo-adjuvant or adjuvant agents with the goal to increase the effectiveness of the primary therapy. Hence, future research is necessary for evaluating the role of antibiotics in preventing breast cancer invasiveness. 

## 4. Material and Methods

### 4.1. Mammalian Cell Culture

The normal and breast cancer epithelial cell lines were grown using standard tissue culture conditions at 37 °C with 5% CO_2_ in humidified environment. MCF 10A (ATCC CRL-3062, Manassas, VA, USA) was cultured in DMEM/Ham’s F-12 (Sigma Aldrich, St. Louis, MO, USA) supplemented with 100 ng/mL cholera toxin (Sigma Aldrich), 20 ng/mL epidermal growth factor (EGF, Sigma Aldrich, St. Louis, MO, USA), 0.01 mg/mL insulin (Sigma Aldrich), 500 ng/mL hydrocortisone (Sigma Aldrich), and 5% chelex-treated horse serum (Gibco, ThermoFisher Scientific, Waltham, MA, USA). MDA-MB-231 (ATCC HTB-26) was grown in Dulbecco’s Modified Eagle Medium (DMEM, Sigma Aldrich) with 10% Fetal Bovine Serum (FBS, Gemini Bio, West Sacramento, CA, USA) and 1% Penicillin-Streptomycin-Glutamine (PSG, ThermoFisher Scientific). 

All infections were carried out in co-culture media composed of the mixture of 2/3rd BSK-H medium (Sigma Aldrich, St. Louis, MO, USA) with 6% rabbit serum and 1/3rd serum and antibiotic-free growth media, depending on the cell type. 

### 4.2. Bacteria Culture Conditions

Low passage isolates (<6) of *B. burgdorferi* B31 strain (ATCC 35210) were cultured in BSK-H media (Sigma Aldrich) supplemented with 6% rabbit serum (Pel-Freeze, Rogers, AR, USA). Cultures were grown in sterile 15 mL glass tubes at 33 °C and 5% CO_2_, and a concentration of 1 × 10^6^ spirochete/mL was used. Cell pellets were then centrifuged at 3000 rpm for 10 min at room temperature (RT). Spirochete pellets were re-suspended into 1 mL of co-culture media for the infection studies.

### 4.3. Immunofluorescence

MCF 10A and MDA-MB-231 cells (4 × 10^4^/well) were grown on sterile 4-well chamber slides (ThermoFisher Scientific) for 24 h. The cells were then washed with 1X phosphate buffer saline (PBS, PH 7.4) after 24 h and were infected with *B. burgdorferi* B31, at a multiplicity of infection (MOI) of 60, resuspended in 1 mL of co-culture media. Uninfected cells grown in co-culture media were used as a negative control. The uninfected and infected normal and breast cancer epithelial cells were incubated for 4, 24, 48, and 72 h at 37 °C, 5% CO_2_. After the infection, the co-culture media was removed, and cells were washed twice with 1× PBS pH 7.4. The infected cells were treated with 10 µg/mL of Cell tracker CM-Dil dye (ThermoFisher Scientific, Waltham, MA, USA, C7001) diluted in 1× PBS pH 7.4 to stain the epithelial cells. The cells were incubated at 37 °C for 5 min and then transferred to 4 °C for 15 min. The dye was then aspirated, and the cells were washed once with 1× PBS pH 7.4, then fixed using cold methanol for 5 min at −20 °C. The methanol was aspirated, and cells were washed once with 1× PBS pH 7.4. The cells were then treated with a 1:200 dilution of a FITC labeled polyclonal rabbit anti-Borrelia burgdorferi antibody (PA-1-73005, Thermo Scientific) to stain *B. burgdorferi* and incubated at 37 °C for 30 min. The cells were then washed 3× with 1× PBS pH 7.4 and a 1:1000 dilution of 4′,6-diamidino-2-phenylindole (DAPI, ThermoFisher Scientific, Waltham, MA, USA) in 1× PBS pH 7.4 was added to the cells to stain the nucleus of the cells. The cells were incubated for 5 min at room temperature (RT), washed once with 1× PBS pH 7.4, and mounted with VECTASHIELD Antifade Mounting Medium (VECTOR Laboratories, Burlingame, CA, USA). Z-stacks of the stained cells were generated at a magnification of 630× using Leica SP8 confocal microscope along with Leica Application Suite X software for image processing (Yale University, West Campus Microscopy Facility, West Haven, CT, USA). 

### 4.4. Cell Viability

MCF 10A and MDA-MB-231 cells (4 × 10^4^/well) were seeded in 48-wells flat-bottom cell culture plates (Corning Inc, Corning, NY, USA) (six samples per condition) and allowed to attach for 24 h at 37 °C, 5% CO_2_. The growth media was removed, and the cells were washed with 1× PBS pH 7.4. The cells were then infected with *B. burgdorferi* B31 spirochetes, at an MOI of 60, resuspended in 1 mL of co-culture media. The microtiter plate was incubated for further 72 h and then cell proliferation was measured using the XTT Cell Proliferation Assay Kit (ATCC, Manassas, VA, USA 30-1011K) according to the manufacturer’s instructions. 

### 4.5. Matrigel Invasion Assay

Cell invasiveness of MCF 10A and MDA-MB-231 cells were analyzed using the trans-well insert membrane system coated with basement membrane matrix, Matrigel (Corning Inc, Corning, NY, USA). 8 μm pore size cell culture inserts (Corning) were placed in a 24-well culture plate (Corning Inc, Corning, NY, USA) and were coated with Matrigel diluted with serum-free media at a concentration of 2 mg/mL for both MCF 10A and MDA-MB-231 cells. The inserts were allowed to dry in the laminar flow hood for up to 3 h and then rehydrated by adding serum-free respective media into the inserts. 200 μL of the appropriate concentration of epithelial breast cells suspended in growth media were added to the upper chamber of the inserts while the 800 μL of growth media was added to the lower chamber. The plates were incubated for 6 h at 37 °C, 5% CO_2_ to allow the cells to attach to the basement membrane of the inserts. Following incubation, the growth media was removed from both chambers of the inserts. 200 μL of *B. burgdorferi* B31, at a MOI of 60, resuspended in co-culture media was added to the upper chamber of the inserts and 800 μL of co-culture media was added to the lower chamber of the insert. For the uninfected control group, co-culture media was added to the upper chamber instead of *B. burgdorferi* B31. The plate was incubated for 72 h at 37 °C with 5% CO_2_, and following the incubation, the non-invading cells were carefully removed from the upper chamber of the inserts by wiping with a sterile cotton swab. The membrane was then fixed and stained using the Kwik Diff Kit (9990700, ThermoFisher Scientific, Waltham, MA, USA). The invasive cells adhering to the lower surface of the membrane were counted under a light microscope. 

### 4.6. Wound-Healing Assay

Cell migration rate of the epithelial breast cells following the infection with *B. burgdorferi* was analyzed using a standard wound-healing assay [21]. The control sample contained the cells that were not infected with *B. burgdorferi* B31 cells. MCF 10A and MDA-MB-231 cells were plated at a concentration of 4 × 10^6^ cells/well in 24-well plates and incubated for about 48 h at 37 °C with 5% CO_2_ to allow the cells to reach a confluence of about 70%. The cells were then infected with *B. burgdorferi* B31, at a MOI of 60, resuspended in 1 mL co-culture media and incubated for 48 h. Following the incubation, the cells were wounded by scratching with a sterile p10 pipette tip as described previously [16]. Thereafter, the media containing the non-attached cells were removed and the cells were washed once with 1 mL of sterile 1× PBS (pH 7.4), followed by adding 1 mL of co-culture media to each well. The cells were incubated at 37 °C, 5% CO_2._ The cell migration was assessed by imaging using ZOE Fluorescent Cell Imager (Bio-Rad, Hercules, CA, USA) and quantifying the gap at 0 and 6 h using Image J software (ImageJ.nih.gov, https://imagej.nih.gov/ij/index.html). 

### 4.7. Western Blot

MDA-MB-231 and MCF 10A (2 × 10^6^ cells) were cultured in 100 mm dishes and infected with *B. burgdorferi* for 24, 48, and 72 h, respectively. The cells were rinsed once with cold 1× PBS pH 7.4, lysed with NP-40 lysis buffer (J60766, Alfa Aesar, Ward Hill, MA, USA), and scraped off using cell scrapers. This lysate was centrifuged at 12,000 rpm for 10 min at 4 °C. The supernatant was collected, and protein content quantified using Bradford’s method and stored at −80 °C until further use. Protein extracts were prepared using 2× Laemmli Buffer (Bio-Rad, Hercules CA, USA) and beta-mercaptoethanol. 35 µg of protein was electrophoresed using 4–15% Mini-PROTEAN TGX gels (Bio-Rad) and 1× Tris Glycine/SDS running buffer (Bio-Rad). The gels were transferred using Trans-Blot Turbo Transfer Pack (Bio-Rad) and Bio-Rad Trans-Blot Turbo System. The membranes were blocked overnight with 1% BSA in 1× TBST (1× Tris Buffered Saline in 0.10% Tween 20) at 4 °C. Subsequently they were incubated for 1 h at 4 °C using the following primary antibodies and dilutions: Fibronectin (1:500, GTX112794, GeneTex, Irvine, CA, USA), E-cadherin (1:3000, GTX100443, GeneTex,), N-cadherin (1:3000, ab18203, Abcam, Cambridge, UK), MMP-9 (1:500, MA5-29732, Invitrogen), Vimentin (1:1000, GTX100619, GeneTex,), Twist ½ (1:1000, GTX127310, GeneTex,) and GAPDH (1:1000, 06/2020, Cell Signaling, Beverly, MA, USA). This was followed by incubation for 1 h at 4 °C with HRP-labelled Rabbit IgG secondary antibody (1:10,000, ab205718, Abcam, Cambridge UK). Between each step, the blots were washed thrice with 1× Tris buffered saline with Tween20 (TBST) on a shaker for 10 min at RT. Further, blots were incubated with 1:1 ECL chemiluminescent substrate and peroxidase solution (#1705060, BioRad) for 5 min and visualized using FluorChem (Protein Simple, San Jose, CA, USA) under chemiluminescence settings. Quantitative densitometric analysis was conducted of the protein bands using ImageJ software (ImageJ.nih.gov, https://imagej.nih.gov/ij/index.html, accessed on 12 October 2021)). Obtained data were normalized to GAPDH housekeeping control.

### 4.8. Statistical Analysis

All experiments were performed at least three times independently and each experiment was done with minimum of four samples (see specific experimental conditions in the Results section for each assay). Statistical analysis was performed using Student T-test of mean comparison using Microsoft Excel (Redmond, WA, USA). Data was normalized to uninfected cells control condition and presented as the mean ± standard error of the mean (SEM).

## 5. Conclusions

Results from this study provided evidence that *B. burgdorferi* can invade breast cancer tumor cells and it can increase their tumorigenic phenotype, which strongly suggests that *B. burgdorferi* can have a role in breast cancer development. 

## Figures and Tables

**Figure 1 antibiotics-10-01295-f001:**
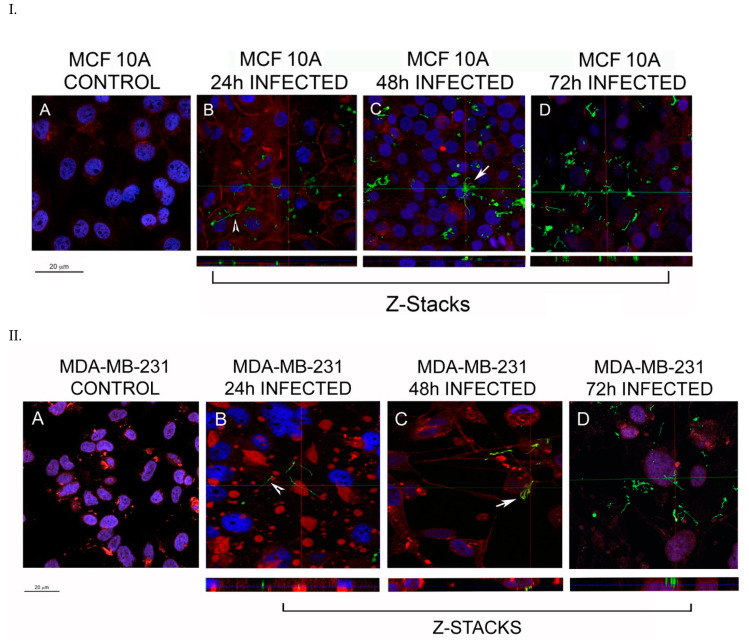
Representative confocal microscopy images of MCF 10A (Panel I) and MDA-MB-231 (Panel II) mammalian epithelial cells infected with *B. burgdorferi*. The cells were co-cultured with *B. burgdorferi* for (**B**) 24 h, (**C**) 48 h, and (**D**) 72 h. (**A**) Uninfected cells treated only with co-culture media were used as a control. CM-Dil membrane stain was used to label the cells (red), DAPI was used to stain the nucleus (blue) and polyclonal anti-*B. burgdorferi* was used to visualize *B. burgdorferi* cells (green), as described in the Material and Methods. The representative images were taken at 630X with a scale bar of 20 μm. White arrowhead represents an invading spirochete and white arrow shows small *B. burgdorferi* aggregates inside the cells.

**Figure 2 antibiotics-10-01295-f002:**
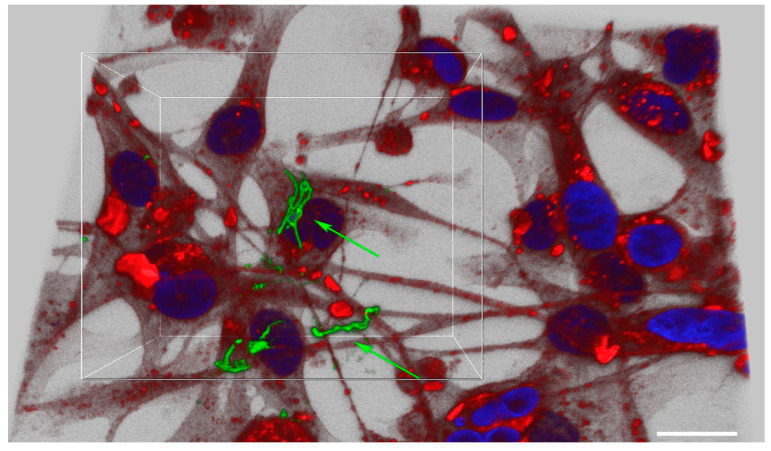
Analyses of *B. burgdorferi* localization in infected MDA-MB-231 mammalian epithelial using confocal microscopy. The cells were co-cultured with *B. burgdorferi* for 48 h. CM-Dil stain was used to label live cells (red), DAPI was used to stain the nucleus (blue) and polyclonal anti-*B. burgdorferi* was used to visualize *B. burgdorferi* (green) as described in the Material and Methods. The image was taken at 630X with a scale bar of 20 μm.

**Figure 3 antibiotics-10-01295-f003:**
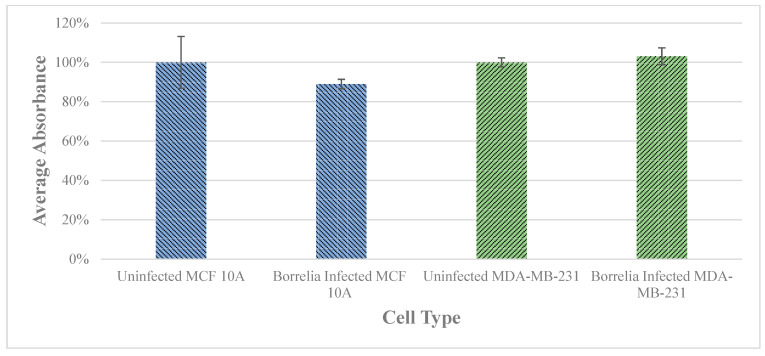
The effect on cell proliferation of MCF 10A an MDA-MB-231 epithelial cells after infection with *B. burgdorferi* versus uninfected breast epithelial cells after 72 h determined by a cell viability assay as described in Material and Methods. The error bar represents a standard error of mean (SEM). All experiments were performed a minimum of three independent times with at least six samples per experiment (*n* = 18).

**Figure 4 antibiotics-10-01295-f004:**
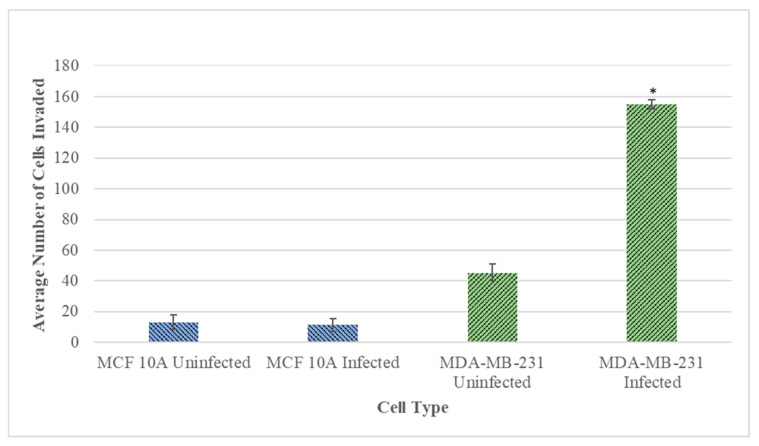
Quantitative analysis of the Matrigel-based invasion assay of normal MCF 10A and neoplastic MDA-MB-231 cells infected with *B. burgdorferi* compared to uninfected cells after 72 h using 2 mg/mL concentration of Matrigel as described in Material and Methods. The *p*-value of the data < 0.05 is considered significant and is indicated by *. The error bar represents a standard error of mean (SEM). All experiments were performed a minimum of three independent times with at least six samples per experiment (*n* = 18).

**Figure 5 antibiotics-10-01295-f005:**
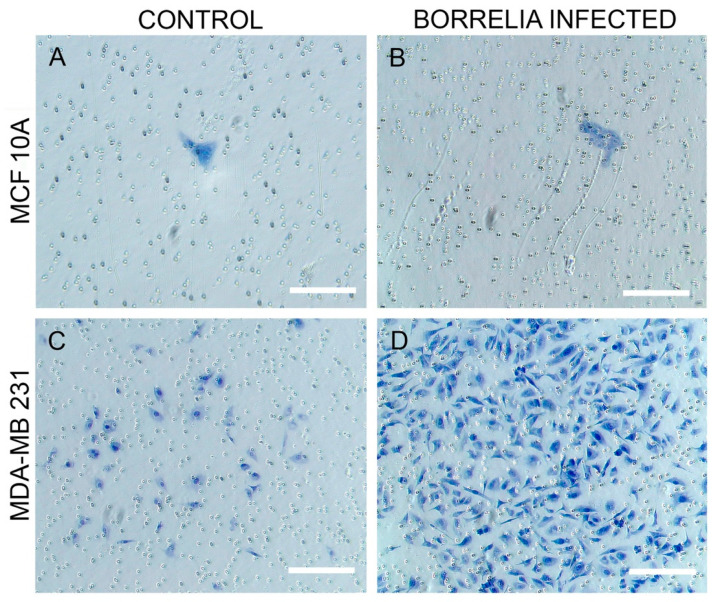
Matrigel-based invasion assay of (**A**) MCF 10A uninfected cells versus (**B**) MCF 10A after infection with *B. burgdorferi*, and (**C**) MDA-MB-231 uninfected cells versus (**D**) MDA-MB-231 after infection with *B. burgdorferi* after 72 h using 2 mg/mL concentration of Matrigel as described in Material and Methods. Scale bar: 200 μm.

**Figure 6 antibiotics-10-01295-f006:**
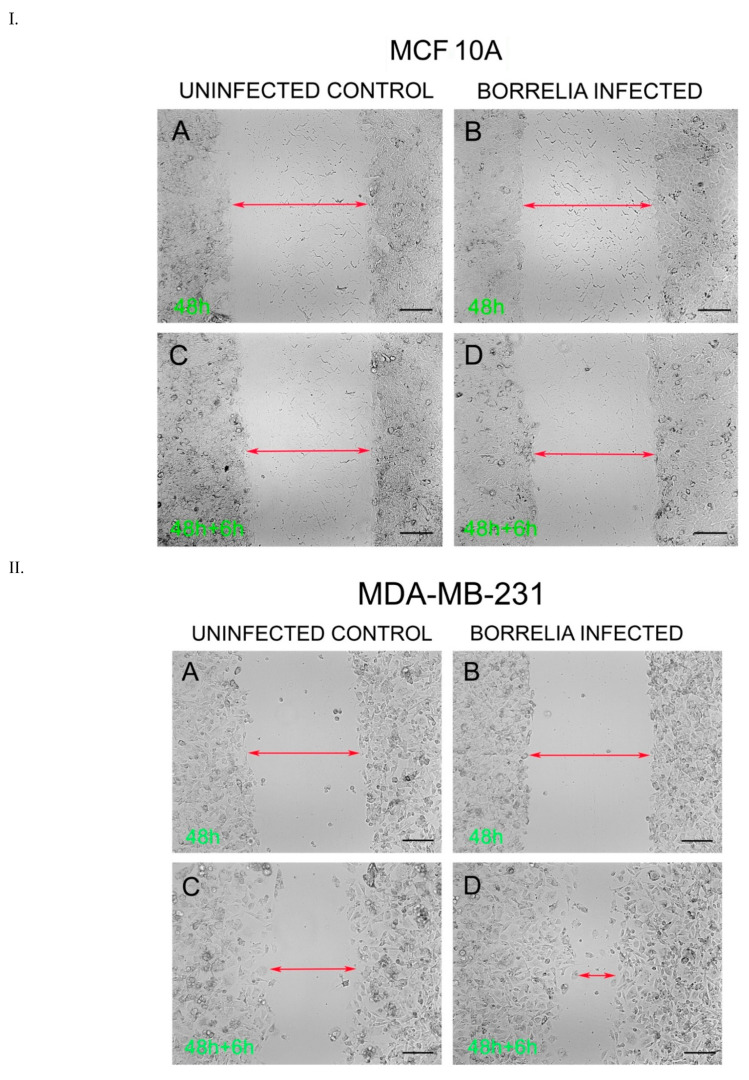
Wound Healing Assay *of B. burgdorferi* infected cells. I. MCF 10A and II. MDA-MB-231 cells after 48 h of infection and 6 h recovery. (Panels **A** and **C**) uninfected versus (Panels **B** and **D**) *B. burgdorferi* infected cells. The cells were infected for 48 h with *B. burgdorferi* and then a wound was induced; cell migration was assessed after an additional 6 h as described in Material and Methods.

**Figure 7 antibiotics-10-01295-f007:**
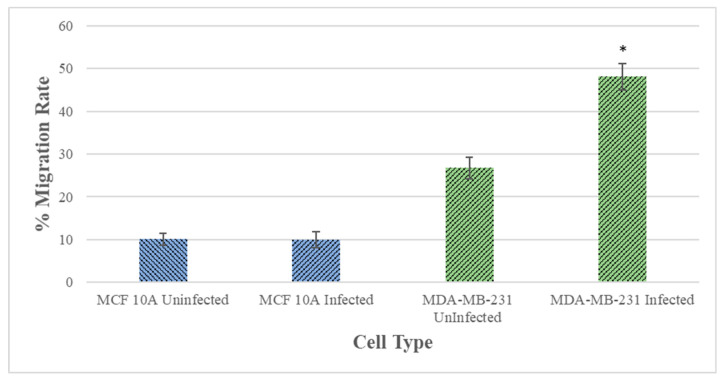
Quantitative analysis of wound healing assay of MCF 10A and MDA-MB-231 epithelial cells after 48 h of infection with *B. burgdorferi* and 6 h of recovery (48 h + 6 h). The cells were infected for 48 h with *B. burgdorferi* and then a wound was induced, and cell migration was quantified after 6 h of recovery as described in Material and Methods. The error bars represent the standard error of mean (SEM). The *p*-value of the data < 0.05 is considered significant and is indicated by *. All experiments were performed a minimum of four independent times with at least three samples per experiment (*n* = 12).

**Figure 8 antibiotics-10-01295-f008:**
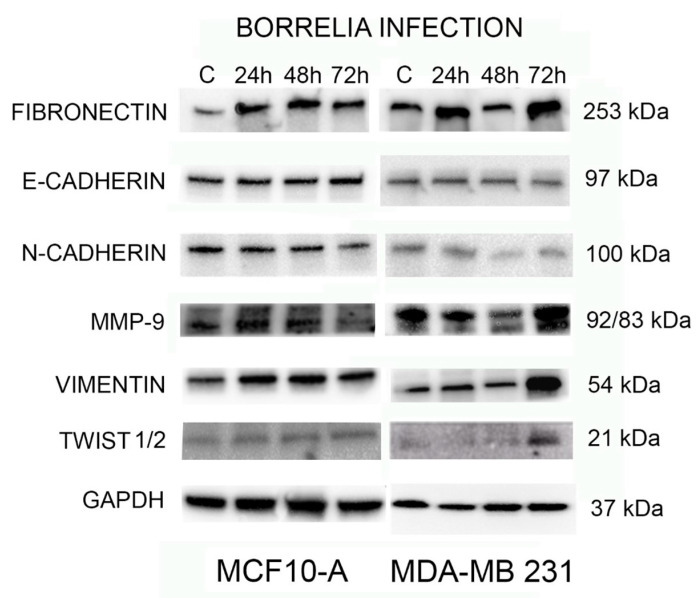
Western blot analysis of expression of EMT-mediating biomarkers and transcription factors: Fibronectin, E-Cadherin, N-Cadherin, MMP-9, Vimentin, and Twist1/2 with loading control GAPDH and uninfected control cells (cultured only in co-infection media) and *B. burgdorferi* infected MCF 10A and MDA-MB-231 cells for 24, 48, and 72 h. A representative blot from three independent experiments was shown.

**Figure 9 antibiotics-10-01295-f009:**
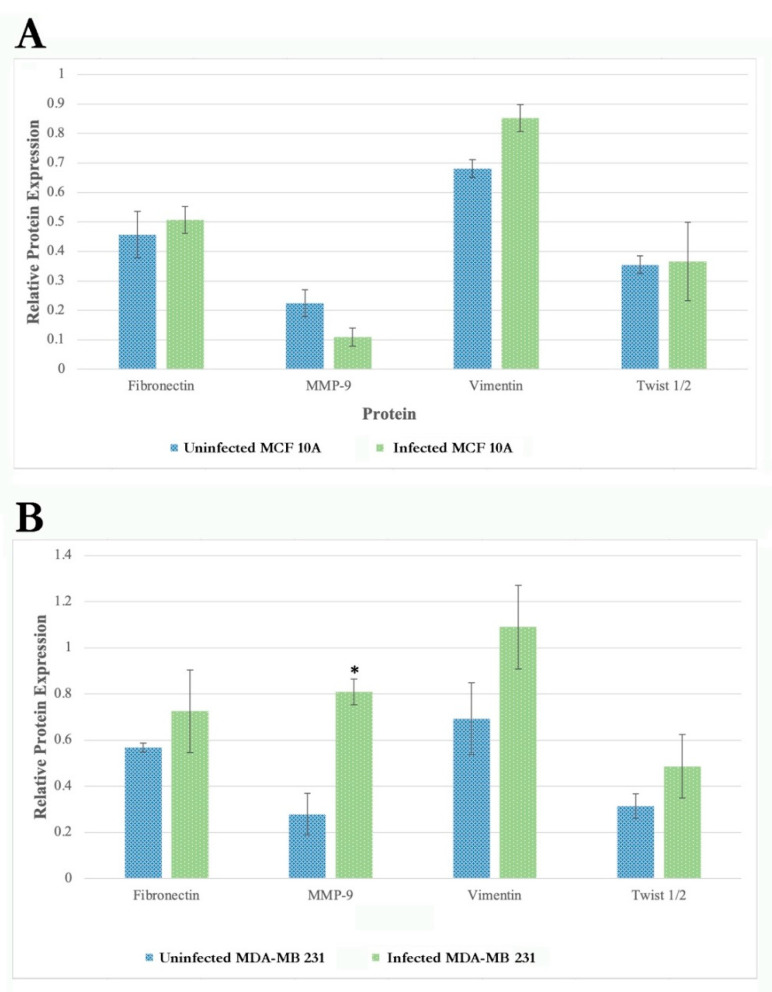
Quantification and statistical analyses of three independent Western blot experiments for the expression levels of Fibronectin, Activated MMP-9, Vimentin, and Twist1/2, as described in Materials and Methods, in uninfected and 72 h infected MCF 10A (Panel **A**) and MDA-MB-231 cells (Panel **B**). Expression levels of the target proteins were normalized for the loading control GAPDH housekeeping protein. Statistical analyses were performed to find significant differences in protein expression level in infected cells compared to uninfected cells as described in Material and Methods. The error bars represent standard error of mean (SEM). The *p*-value of the data < 0.05 is considered significant and is indicated by *.

**Table 1 antibiotics-10-01295-t001:** Quantitative analysis of the number of the different mammary epithelial cell lines invaded by *B. burgdorferi* at different time intervals.

Time	Average Percentage of Cells Invaded by *B. burgdorferi* ± SEM	
	MCF 10A	MDA-MB-231
24 h	36% ± 1%	89% ± 2%
48 h	52% ± 2%	92% ± 1%
72 h	61% ± 1%	96% ± 2%

## Data Availability

The datasets used and/or analyzed during the current study are available from Eva Sapi, University of New Haven.

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
