# Peer review of "Effect of Invasion of Borrelia burgdorferi in Normal and Neoplastic Mammary Epithelial Cells"

_antibiotics, 2021, doi:10.3390/antibiotics10111295_

Round 1
Reviewer 1 Report
The presented work is very interesting. The topic related to B. burgdorferi
infections is hot all over the world. More and more infections are recorded
every year. Can they contribute to the development of neoplastic disease.
This is a problem to be solved. The introduction guides the reader well to the subject. The materials and
methods are clearly described. And a well-written discussion.
But ... is its subject related to the world of antibiotics?
Authors should perform Western blot analysis and try to show which virulence factor is responsible for the ability to invade breast cancer cells
The studies should be compared with another wild-type strain of B. burgdorferi and another cell line. Unless we assume that these are preliminary research (I assumed so at the beginning) and the topic will be developed in the form of another manuscript.
Author Response
We would like to thank our reviewer for the detailed and constructive review of our manuscript. We have incorporated the suggested revisions and updated the manuscript to address the comments/suggestions. We have made every attempt to address all concerns and suggestions in order to make this manuscript ready for publication.
Our responses to our reviewer marked red color.
The presented work is very interesting. The topic related to B. burgdorferi
infections is hot all over the world. More and more infections are recorded
every year. Can they contribute to the development of neoplastic disease.
This is a problem to be solved. The introduction guides the reader well to the subject. The materials and
methods are clearly described. And a well-written discussion.
But ... is its subject related to the world of antibiotics?
Thank you for finding our study interesting, important, and well-written. To answer your question about how this study is related to antibiotics: If we can prove further that Borrelia indeed can be present in metastatic tumors and could have an effect on the invasion of breast cancer cells, it might be possible to use antibiotics as neo-adjuvant or adjuvant therapy to increase the effectiveness of the primary therapy. We added these sentences to the end of the discussion section of the manuscript to emphasize the importance of this question. There are two very interesting studies in which the researchers evaluated antibiotics used for treating Lyme disease for cancer treatment (De Francesco EM et al and Dong XZ et al)
- De Francesco E.M. et al. (2017) Targeting hypoxic cancer stem cells (CSCs) with doxycycline: Implications for optimizing anti-angiogenic therapy. Oncotarget 8(34), 56126-56142. doi:10.18632/oncotarget.18445
- Dong Z. et al. (2019) Biological functions and molecular mechanisms of antibiotic tigecycline in the treatment of cancers. Int J Mol Sci 20(14), 3577. doi: 3390/ijms20143577.
Authors should perform Western blot analysis and try to show which virulence factor is responsible for the ability to invade breast cancer cells.
We are planning to address this question in our future studies by using mutant cells for important virulence factors such as fibronectin binding protein (bbk32), decorin binding protein A (dbpBA), outer surface protein A (ospA), and surface-localized lipoprotein (bba64) mutants based on the data from previous studies on important factors for host cell invasion (Schmit VL et al and Hyde JA). We are also planning to use specific antibodies for those virulence proteins and treat wild type Borrelia burgdorferi to evaluate which proteins can block the invasion by Borrelia burgdorferi into epithelial cells.
Schmit VL, Patton TG and Gilmore RD Jr. (2011) Analysis of Borrelia burgdorferi surface proteins as determinants in establishing host cell interactions. Front Microbio 2:141. doi: 10.3389/fmicb.2011.00141
Hyde JA (2017) Borrelia burgdorferi Keeps Moving and Carries on: A Review of Borrelial Dissemination and Invasion. Front Immunol 8:114. doi: 10.3389/fimmu.2017.00114
The studies should be compared with another wild-type strain of B. burgdorferi and another cell line. Unless we assume that these are preliminary research (I assumed so at the beginning) and the topic will be developed in the form of another manuscript.
Yes, we are also planning extend this study and use several wild-type B. burgdorferi sensu lato strains to evaluate any differences in the invasion capacity (B. burgdorferi 297 and N40 as well as European strains Borrelia afzelii and garinii). For mammalian cells, we are continuing the study with various breast cancer cells with different hormonal receptor and HER-2/Neu status such as BT20, T47D, MCF7, SKBR3 cell lines, to better understand the host-pathogen interaction between those mammalian cells and B. burgdorferi.
Thank you again for your great suggestions, it really helped us to validate our future direction with this study.
Reviewer 2 Report
In the manuscript entitled “Effect of invasion of Borrelia burgdorferi in normal and neoplastic mammary epithelial cells” the authors investigated the host-pathogen interactions involving B. burgdorferi and normal and neoplastic mammary epithelial cells. The suggestions to improve the manuscript are given below.
- The study does not address cause/effect relationship between B. burgdorferi and breast cancer. Does B. burgdorferi capable of causing breast cancer? What is the link between Lyme disease and breast cancer? Why B. burgdorferi is only linked to breast cancer? The authors should address these questions in the discussion.
- The authors should do densidometric analysis of the Western blotting data and represent the data quantitatively.
- How many times each experiment was repeated? The results should be expressed as mean +/- standard error of the mean (SEM).
Author Response
We would like to thank our reviewer for the detailed and constructive review of our manuscript. We have incorporated the suggested revisions and updated the manuscript to address the comments/suggestions. We have made every attempt to address all concerns and suggestions in order to make this manuscript ready for publication.
Our responses to our reviewer marked with red color:
In the manuscript entitled “Effect of invasion of Borrelia burgdorferi in normal and neoplastic mammary epithelial cells” the authors investigated the host-pathogen interactions involving B. burgdorferi and normal and neoplastic mammary epithelial cells. The suggestions to improve the manuscript are given below.
- The study does not address cause/effect relationship between B. burgdorferi and breast cancer. Does B. burgdorferi capable of causing breast cancer? What is the link between Lyme disease and breast cancer? Why B. burgdorferi is only linked to breast cancer? The authors should address these questions in the discussion.
Thank you for the great suggestions. The reason we did not discuss the potential relationship between B. burgdorferi infection and breast cancer development is because we are in the process of evaluating the presence of B. burgdorferi in different breast cancer tissues. While we already have some promising preliminary data, we felt the need to complete the project before we can state a potential cause/effect relationship. But we agree that we should at least mention the potential role of B. burgdorferi in cancer development in the discussion, therefore we added a few sentences to the end of the discussion.
- The authors should do densitometric analysis of the Western blotting data and represent the data quantitatively.
We did perform densitometric analysis to quantify the Western data, but we did not use extensive statistical analyses and we agree with our reviewer that it is very important, Therefore, we reanalyzed the data carefully and included the results as a new graph in Figure 9.
We especially want to say thank you for this great suggestion, as additional analyses revealed a very significant result for activated MMP9 expression in infected breast cancer cells. The corresponding result sections as well as the abstract were edited according to this new finding.
How many times each experiment was repeated? The results should be expressed as mean +/- standard error of the mean (SEM).
All experiments were performed at least 3 times and each experiment was done with minimum 4 samples (added to the Material and Methods sections and added to all corresponding figure legends). The SEM was calculated using the following formula in Microsoft Excel: =STDEV(sampling range)/SQRT(Count of sampling range) and all quantitative data are now shown with SEM.